# The Spectrum of Invasive Fungal Sinusitis in COVID-19 Patients: Experience from a Tertiary Care Referral Center in Northern India

**DOI:** 10.3390/jof8030223

**Published:** 2022-02-24

**Authors:** Surendra Singh Baghel, Amit Kumar Keshri, Prabhakar Mishra, Rungmei Marak, Ravi Sankar Manogaran, Pawan Kumar Verma, Arun Kumar Srivastava, Raj Kumar, Arulalan Mathialagan, Govind Bhuskute, Abhishek Kumar Dubey, Radha Krishan Dhiman

**Affiliations:** 1Neurootology Unit, Department of Neurosurgery, Sanjay Gandhi Postgraduate Institute of Medical Sciences (SGPGIMS), Lucknow 226014, India; surendrabaghel3@gmail.com (S.S.B.); amitkeshri2000@yahoo.com (A.K.K.); arulalan87@gmail.com (A.M.); gowind.bhuskute@gmail.com (G.B.); entdubey@gmail.com (A.K.D.); 2Department of Biostatistics and Health Informatics, Sanjay Gandhi Postgraduate Institute of Medical Sciences (SGPGIMS), Lucknow 226001, India; mishrapk79@gmail.com; 3Department of Microbiology, Sanjay Gandhi Postgraduate Institute of Medical Sciences (SGPGIMS), Lucknow 226001, India; rungmei@gmail.com; 4Department of Neurosurgery, Sanjay Gandhi Postgraduate Institute of Medical Sciences (SGPGIMS), Lucknow 226014, India; dr.pawankverma@gmail.com (P.K.V.); doctorarunsrivastava@gmail.com (A.K.S.); rajkumar1959@gmail.com (R.K.); 5Head Department of Hepatology, Sanjay Gandhi Postgraduate Institute of Medical Sciences (SGPGIMS), Lucknow 226014, India; rkpsdhiman@gmail.com

**Keywords:** invasive fungal sinusitis, glycemic control, steroid use, overall survival rate

## Abstract

This study aimed to determine the patient demographics, risk factors, which include comorbidities, medications used to treat COVID-19, and presenting symptoms and signs, and the management outcome of COVID-19-associated invasive fungal sinusitis. A retrospective, propensity score-matched, comparative study was conducted at a tertiary care center, involving 124 patients with invasive fungal sinusitis admitted between April 2021 and September 2021, suffering from or having a history of COVID-19 infection. Among the 124 patients, 87 were male, and 37 were female. A total of 72.6% of patients received steroids, while 73.4% received antibiotics, and 55.6% received oxygen during COVID-19 management. The most common comorbidities were diabetes mellitus (83.9%) and hypertension (30.6%). A total of 92.2% had mucor, 16.9% had aspergillus, 12.9% had both, and one patient had hyalohyphomycosis on fungal smear and culture. The comparative study showed the significant role of serum ferritin, glycemic control, steroid use, and duration in COVID-19-associated invasive fungal disease (*p* < 0.001). Headache and facial pain (68, 54.8%) were the most common symptoms. The most involved sinonasal site was the maxillary sinus (90, 72.6%). The overall survival rate at the three-month follow-up was 79.9%. COVID-19-related aggressive inflammatory response, uncontrolled glycemic level, and rampant use of steroids are the most important predisposing factors in developing COVID-19-associated invasive fungal sinusitis.

## 1. Introduction

Coronavirus disease 2019 (COVID-19) is an infection caused by severe acute respiratory syndrome coronavirus-2 (SARS-CoV-2). Since the first case was detected in December 2019 in Wuhan, China, there have been various twists and turns in the understanding of its pathophysiology, diagnosis, management, sequelae, and complications [1]. COVID-19 has been associated with a wide range of opportunistic infections [2]. Recent observations have revealed a dangerous and potentially deadly complication, invasive fungal sinusitis, resulting specifically from mucormycosis. Sinonasal mucormycosis is a life-threatening invasive fungal sinusitis that typically affects immunocompromised individuals. The high-risk groups are those with uncontrolled diabetes mellitus, acquired immunodeficiency syndrome, iatrogenic immunosuppression, and hematologic malignancies, and those who have undergone organ transplantation [3,4,5].

Clinically, invasive fungal sinusitis can present with atypical signs and symptoms similar to complicated sinusitis [6,7]. Invasive fungal sinusitis refers to the entire spectrum ranging from limited sinonasal disease (sinonasal tissue invasion), sinonasal palatal disease (progression to the palate), rhino-orbital disease (progression to orbits) to rhino-orbital-cerebral disease (CNS involvement). In addition, there may be a rapid disease progression without early diagnosis and treatment, with a reported survival rate ranging from 20% to 70% [8]. Recently, a surge in the incidence of mucormycosis infection of the sinuses has been observed, with more cases being diagnosed much more frequently. A complex interplay of factors, including diabetes mellitus, existing respiratory pathology, immunosuppression therapy, nosocomial infection, and immune alterations by COVID-19 infection, are associated with its pathogenesis [9,10].

Furthermore, COVID-19 possibly plays an essential role because affected patients show overexpression of inflammatory cytokines and impaired cell-mediated immunity with a decreased cluster of differentiation of 4 and 8 positive T-helper (CD4+ T and CD8+ T) cell counts. This indicates susceptibility to fungal coinfection [10,11], along with high blood glucose (diabetes, new-onset hyperglycemia, steroid-induced hyperglycemia), low oxygen (hypoxia), acidic medium (metabolic acidosis, diabetic ketoacidosis (DKA), high iron levels (increased ferritin), decreased phagocytic activity of white blood cells (WBC), and immunosuppression (SARS-CoV-2 mediated, steroid-mediated, immunomodulator-mediated, or background comorbidities), coupled with several other shared risk factors, including prolonged hospitalization with or without mechanical ventilators, increasing the chance of opportunistic fungal infections to colonize [10]. The present study was conducted to study the epidemiological pattern and clinical features of COVID-19 patients with invasive fungal sinusitis. We also compared these patients with propensity-matched COVID-19 patients to identify the risk factors in developing invasive fungal sinusitis.

## 2. Materials and Methods

A retrospective, hospital-based, observational study was conducted at SGPGIMS, Lucknow Tertiary Referral Center, India, over six months, from April 2021 to September 2021. All patients with invasive fungal disease of the paranasal sinuses, either COVID-19-positive or recovered from COVID-19 infection, were included in the study. A control group was formed by collecting data on COVID-19 patients admitted at our center during the same duration. From these control patients, 124 propensity score-matched (age, sex, respiratory support, and comorbidities) patients were identified among those who did not develop invasive fungal sinusitis. Patient age, gender, history, COVID-19 management details, clinical presentation, imaging findings, comorbidities, pathology, and follow-up information was obtained, recorded, and analyzed.

**Statistical Analysis:** Continuous variables were represented as mean ± standard deviation and median and interquartile range and categorical variables in frequency (%). The Mann–Whitney U-test was used to compare medians, whereas, for proportions, the Chi-square test was used between the two groups. Propensity score matching was performed in COVID-19 patients with and without invasive fungal sinusitis, where age, sex, type of respiratory support (room air, oxygen, mechanical ventilator), and comorbidities (hypertension and diabetes) were matched. Binary logistic regression analysis was used to assess the independent predictors (from treatment and inflammatory variables) of the invasive fungal sinusitis compared with COVID-19 patients who did not develop invasive fungal sinusitis. A value of *p* < 0.05 was considered statistically significant. Statistical analyses were performed using SPSS version 23.0 software (SPSS Inc., Chicago, IL, USA).

## 3. Results

A total of 124 patients were diagnosed with invasive fungal sinusitis with a history of COVID-19 infection; of them, 87 were male, and 37 were female. The mean age of patients was 51.7 ± 11.54 years (range: 28–82). Twenty patients were COVID-19 positive at the time of presentation, and the remaining 104 had been documented with infection earlier and had recovered. Among the 124 patients, only 66 (53.2%) had been admitted to the hospital for treatment of COVID-19, while the remaining 58 (46.8%) were in home isolation. During management of COVID-19, 90 (72.6%) patients received corticosteroids for a mean duration 11.3 ± 6.6 days (range: 3–45, 91 (73.4%) received antibiotics (48 (52.7%) intravenous antibiotics and 43 (47.3%) oral antibiotics), and 69 (55.6%) were on oxygen support (mask and nasal prong, 44 (63.8%); NRBM, 9 (13%); BIPAP, 9 (13%); mechanical ventilation, 7 (10.1%)) for a mean duration of 9.5 ± 8.13 days (range: 0–45). Tocilizumab was used in 3 (2.4%) patients. The mean duration between becoming COVID-19 positive and the appearance of the first symptom of invasive fungal sinusitis was 35.4 ± 37.15 days (range: 6–292). Fungal sinusitis confirmation was performed by KOH mount, fungal culture, and histopathological examination. On evaluating the type of fungi, we found that 92.1% had mucor, 16.9% had aspergillus, 12.9% had both, and one patient (0.8%) had hyalohyphomycosis. Serum ferritin (performed at time of hospitalization) was elevated in 78.3% (97/124) of patients, with a mean value of 1780.3 ± 7885.63 (Table 1).

The most commonly associated comorbidity was diabetes mellitus (DM) (83.9%). Among the 104 patients diagnosed with diabetes, 9 (8.6%) had a history of a recent onset, probably caused by corticosteroid therapy given during COVID-19 treatment. Of the 104 patients, 88 (84.6) had uncontrolled DM, and 16 (15.4) had controlled DM based on HbA1c values. The mean HbA1c was 9.1 ± 2.26. Other comorbidities in our study populations were hypertension seen in 38 (30.6%), heart disease in 8 (6.5%), liver disease in 5 (4%), hypothyroidism in 5 (4%), and kidney disease among 16 (12.9%) and 5 (4%) was post-renal transplant patients on immunosuppressive drugs. One patient was HBsAg positive, one was HCV positive, and one had acute chronic pancreatitis. Three patients had concomitant pulmonary and rhino-orbital mucormycosis (Table 2).

The “classic” triad of DM, steroid therapy, and oxygen use was present in 54 (43.5%) patients, while 29 (23.4%) had only DM, and 5 (4%) received only steroids. No patient developed invasive fungal sinusitis with oxygen treatment alone. Four patients were nondiabetic and did not use steroids and oxygen during treatment of COVID-19. Of these four, one patient was a post-renal transplant and was on immunosuppressant therapy, and the other three did not have any comorbidities.

We compared our data with the same number of COVID-19 patients’ data with the same age, sex, oxygen support, and comorbidities (DM, HTN) with those who did not develop mucor. Comparative univariate analysis showed that serum ferritin (median: 576 vs. 244, *p* < 0.001), HbA1c (median: 9.1 vs. 7.7, *p* < 0.001), proportion of patients receiving steroid (90% vs. 39%, *p* < 0.001), and duration of steroid use (median days: 10 vs. 7, *p* < 0.001) were significantly higher in invasive fungal sinusitis patients compared to control patients. Multivariate binary logistic regression was used to calculate the adjusted odds ratio (AOR) after including significant variables from univariate analysis. In this analysis, except the use of steroids, the other variables, i.e., serum ferritin (AOR: 1.001, *p* = 0.041), HbA1c (AOR: 1.28, *p* < 0.001), and duration of steroid use (AOR: 1.12, *p* = 0.014), were significant and found to be independent risk factors in development of invasive fungal sinusitis (Table 3).

Contrast-enhanced MRI is the imaging modality of choice, as it clearly delineates the soft-tissue involvement earlier and is better than a CT scan, especially in orbital and intracranial involvement. A contrast-enhanced CT scan is relatively faster and can be used for patients where MRI is not feasible. CT provides better bony details. Mucormycosis leads to soft-tissue necrosis due to vasoinvasion, leading to vascular compromise and bone erosion not being a common finding, so a CT scan may not identify early disease. MRI was performed in 107 cases. Among the remaining 17 patients for whom MRI could not be performed, 1 had a history of a gunshot, 1 had minimal palatal involvement, and the rest were on ventilator support.

The most commonly involved sinonasal site was the maxillary sinus, seen in 90 (72.6%) patients, followed by ethmoid sinuses in 87 (70.2%) patients. The lateral nasal wall was involved in 77 (62.1%) patients, and the sphenoid sinus was involved in 56 (45.2%) patients. Frontal sinus involvement was less common, seen only in 25 (20.2%) patients. Pterygopalatine fossa and infratemporal fossa were involved in 46 (37.1%) and 38 (30.6%) cases, respectively. Orbital involvement was seen in 44 (35.5%) cases, and the palate was involved in 41 (33.1%) cases, while 23 (18.54%) patients showed intracranial extension, with the spectrum containing five frontal lobe abscesses, three temporal lobe abscesses, two parietal lobe abscesses, nine cavernous sinus thrombosis, three internal carotid artery (ICA) thrombosis, and three fungal cerebritis. Nineteen patients had skull base osteomyelitis (Table 2).

Among the 124 patients, 110 were treated with combined antifungal therapy and surgery, while 14 patients were on antifungal therapy alone without surgical debridement. The reasons for avoiding surgery in these 14 patients were that 2 patients did not consent to surgery, 10 patients were COVID-19 positive (risks outweighed benefits), while the other two were admitted to ICU with severe sepsis, pneumonia, kidney failure, and overall poor general condition, making surgical intervention too unreasonable a risk. Among the 110 patients undergoing surgery, 93 were primary cases, and 17 were revisions. Of the 17 patients, 14 underwent revision surgery, and only antifungals were used to manage 3 patients. Only one patient underwent surgery in the RT-PCR-positive status; the rest underwent surgery after two consecutive negative COVID-19 RT-PCR reports. Orbital exenteration was performed in 5 patients. Intracranial abscess drainage was performed in 5 patients.

Amphotericin B (deoxycholate, liposomal, lipid complex) was given as the primary antifungal drug in 116 patients, and isavuconazole was given in 8 patients. Posaconazole and voriconazole were used as step-down therapy for three months according to histopathology report and microscopy report.

The overall survival rate was 79.8% (99/124) in our study after a mean follow-up of three months. Of the 25 (19.4%) patients who expired, 11 had an active COVID-19 infection at the time of death. Finally, 13 died due to severe respiratory failure and sepsis due to COVID-19, 9 had intracranial extension (5 intracranial abscess, 2 ICA thrombosis, and 2 cavernous sinus thrombosis), and 2 patients had simultaneous pulmonary mucormycosis. Three patients were post-renal transplants and were on triple immunosuppressants. Four patients developed hemiparesis in the immediate post-op period.

## 4. Discussion

Mucormycosis or zygomycosis, also called phycomycosis, initially described in 1885 by Paltauf, is an uncommon and aggressive fungal infection that usually affects patients with alteration of their immunological system [12]. Although it has a low incidence rate, varying from 0.005 to 1.7 per million population, many cases have been seen recently, amounting to a significant increase in its incidence in the coronavirus pandemic [13,14,15]. It can affect the nose, sinus, orbit, central nervous system (CNS), lung, gastrointestinal tract, skin, jawbones, heart, kidney, and mediastinum. It is a lethal fungal disease, with rhinocerebral presentation being its most common form [16].

### 4.1. Epidemiology

In our study population, the mean age of COVID-19 patients admitted to the hospital was 48.24 years (range: 36–61), and 71.8% of patients were males [17]. The median age of affliction of COVID-19-associated rhino-orbital-cerebral mucormycosis (ROCM) patients has been reported to be 51.9 years (range: 12–88), and there was a male predilection (71%) [18]. The demographic profile in our series was consistent with these studies, with a mean age of 51.7 (range: 28–82) years and 70.2% male patients. Greater outdoor exposure in males may be the possible reason for this majority.

### 4.2. Potential Risk Factors

The etiology of COVID-19-associated invasive fungal sinusitis appears multifactorial. In our population with COVID-19, DM was seen in 31.7% of hospitalized patients. New-onset DM was seen in 20.6% of patients with mild to moderate COVID-19 [19]. Injudicious use of corticosteroids precipitates hyperglycemia, and the virus itself damages the pancreatic islet cells, producing new-onset DM or worsening the pre-existing DM. The cytokine storm indirectly precipitates this by resulting in insulin resistance [20,21,22]. Hyperglycemia leads to glycosylation of transferrin and ferritin and reduces iron binding. By reducing the ability of transferrin to chelate iron, acidosis presents an additive effect, causing an overall increase in free iron levels, allowing mucor to thrive [10].

In the studies of Kursun et al. [23], Mohammadi et al. [24], and Kermani et al. [25], the most common underlying disorders affecting the immune system were diabetes mellitus and, to a lesser extent, hematologic malignancies and chronic renal insufficiency. In an extensive series of cases in the pre-COVID-19 era, 74% of patients had diabetes [26]. People with diabetes had 7.5 times higher chances of developing mucormycosis than the general population [27]. In the current COVID-19-associated ROCM in India, in an extensive study of 2826 cases by Sen et al. [18], 82% were diabetic. The literature review of the existing global data by Singh et al. [10] showed that people with diabetes account for 80% of cases. Our data showed that 83.9% (104/124) were diabetic, 88/104 (84.6%) had uncontrolled DM (HbA1c > 7), and 78.3% (97/124) had elevated serum ferritin levels. Comparing the data with the control group after propensity matching (age, sex, DM, HTN, oxygen support) showed COVID-19-related aggressive inflammatory response (increased serum ferritin levels) and chronically uncontrolled blood sugar levels (elevated HbA1c). Use of high-dose steroid and steroid use for a prolonged duration leads to an increased chance of developing invasive fungal disease (Table 3).

Extensive use of steroids and broad-spectrum antibiotics in the management of COVID-19 leads to the development or exacerbation of a pre-existing fungal disease [28]. According to the Randomized Evaluation of COVID-19 Therapy (RECOVERY) collaborative group, the National Institute of Health recommends steroid use only in patients on a ventilator or who require supplemental oxygen, but not in milder disease [29]. The updated national guidelines recommend the use of intravenous methylprednisolone (or an equivalent of 0.1–0.2 mg/kg dexamethasone) at a dose of 0.5–1 mg/kg in two divided doses for moderate disease and 1–2 mg/kg in two divided doses (0.2–0.4 mg/kg dexamethasone) for severe disease for 5–10 days [30]. Based on the literature review of global data, 76% of patients with COVID-19-associated ROCM gave a history of systemic corticosteroids use [10]. In India, this fraction is higher, closer to 87% [18]. Our comparative study showed that 72% of patients used steroids in the mucor group, while in the control group, only 31.5% of patients used steroids and the average duration of steroid use was 11.3 days in the mucor group, and in the control group, it was 8 days. Thus, injudicious and prolonged use of corticosteroids, especially in immunocompromised patients, i.e., DM, can increase the chance of acquiring invasive fungal sinusitis.

The mode of contamination occurs through the inhalation of fungal spores. Our data of COVID-19-associated invasive fungal sinusitis show that 55.6% (69 of 124) patients required oxygen support during the treatment of COVID-19, but, at the same time, either they had DM or had received corticosteroids in some form. No patient in our study who received only oxygen therapy as a treatment modality for COVID-19 developed invasive fungal sinusitis, suggesting that contaminated oxygen is unlikely to be a cause of infection. In addition, four (3.2%) patients were nondiabetic and did not receive oxygen and steroid; three (75%) out of four were immunocompetent but still developed mucormycosis, raising a possible role of COVID-19 as a culprit. Further, an ubiquitous, fungi-like mucor, coupled with poor hygienic practices followed at the individual and community levels, can be speculated as one of the causes for such high incidence [31].

### 4.3. Timing of Occurrence

Invasive fungal sinusitis occurs both concomitantly with COVID-19, as well as post-recovery. In a large multicenter study of 2285 patients by Sen et al. [18], 56% of patients developed mucormycosis symptoms within 14 days from the diagnosis of COVID-19. Additional spikes were seen on Days 15 and 20 with 10% and 7% of patients, respectively, while 44% of patients presented with symptoms of mucormycosis following recovery from COVID-19. Delayed mucormycosis, three months post-COVID-19, was noted in seven patients.

In our study, the mean duration between COVID-19-positive RT-PCR and the first symptom of invasive fungal sinusitis was 35.4 ± 37.15 days (range: 6–292). In our study, 4.03% developed signs and symptoms in 7 days, 22.58% developed between 8 and 15 days, 32.2% developed between 16 and 30 days, and 33.06% developed between 30 and 90 days from the diagnosis of COVID-19. Only 8.06% developed symptoms after 90 days. This revealed sinusitis as a red flag sign in the immediate post-COVID-19 period, specifically in high-risk patients. This proposed need for early ENT examination and emphasized the education of patients and families with a checklist of symptoms of invasive fungal sinusitis because 92% of patients developed symptoms within three months after recovery from COVID-19. It has been observed that patients with early diagnosis and limited disease extension have the best outcome with minimal morbidity and mortality [32]. The same outcomes were seen in our study. In addition, 16.1% of patients were positive at the time of diagnosis, which proposed significant challenges in management such as termination of steroids, surgical fitness, and performing surgery wearing a personal protective equipment (PPE) kit.

Clinically, invasive fungal sinusitis can present with atypical signs and symptoms similar to complicated sinusitis, such as headache, nasal blockage, nasal crusting, proptosis, facial pain and edema, ptosis, chemosis, and even ophthalmoplegia, with headache and fever and various neurological signs and symptoms such as lower cranial nerve palsy altering sensorium in patients with intracranial extension [6,7]. In our study, headache and facial pain were seen in 68 (54.8%) patients, facial or periorbital swelling in 50 (40.3%), tooth pain and loosening in 33 (26.6), visual loss in 30 (24.2%), facial numbness in 21 (16.9%), nasal discharge in 20 (16.1%), and ophthalmoplegia and ptosis in 21 (16.9%) and were the most common symptoms and signs. Thus, a high index of suspicion in post-COVID-19 patients presenting with complaints of sinusitis is the key to early diagnosis and better management.

A study by El-Kholy et al. showed that the most involved sinonasal site was the lateral nasal wall (86.1%), with the ethmoid (72.2%) and sphenoid (55.6%) sinuses being next. In this study, 27.8% showed extension intracranially, and 33.3% showed palatal involvement [32]. A large multicentric study showed bilateral paranasal sinus involvement in 40% of cases. Diffuse PNS involvement was present in 58% of cases, and the most common involved sinus was maxillary in 32%, followed by ethmoid in 21%. In our study, the most commonly involved sinonasal site was the maxillary sinus in 90 patients (72.6%), followed by the ethmoid 87 (70.2%), lateral nasal wall 77 (62.1%), and sphenoid 56 (45.2%). Only 12.9% of cases showed bilateral PNS involvement. Orbital involvement was seen in 44 (35.5%), and the palate was involved in 41 (33.1%) cases. A total of 23 (18.54%) patients showed intracranial extension, with the cavernous sinus (10/23) being the most common site of intracranial involvement.

### 4.4. Management

As soon as the patient presented with symptoms suggestive of invasive fungal sinusitis, our protocol was to get a fresh contrast-enhanced MRI and CECT of the nose and PNS to know the extent of the disease and for surgical planning. We also performed a diagnostic nasal endoscopy and sent nasal swabs for fungal microscopy and culture. The MRI shows better soft-tissue delineation and better characteristic findings, but the CT provides better bony details. Diagnostic nasal endoscopy and nasal swabs were not always helpful, but they could be performed by the bedside and without any delay when needed. Complete blood count (CBC), liver function test (LFT), renal function test (RFT), blood coagulation profile, HbA1c, an inflammatory marker, COVID-19 RT-PCR test, blood sugar charting, and all investigations for surgical fitness were sent immediately. Management for high blood sugars was started, and after checking RFT, we started antifungals accordingly with strict monitoring. Repeat RFT, LFT, and serum electrolyte were performed daily, and medications were tailored to any changes. As soon as the patient was considered fit for surgery, they were taken for debridement, and antifungals were continued in the post-op period according to the extent of involvement of the disease.

Medical management was with liposomal amphotericin B in 5–10 mg/kg/day doses up to a total cumulative dose of 3 to 5 g depending on the extent of the disease and patient tolerance. If the patient had a hypersensitivity reaction or RFT was deranged, 200 mg isavuconazole TDS was given for two days, followed by 200 mg once daily. For patients with orbital involvement with intact vision or who did not give consent to orbital exenteration, multiple transcutaneous retrobulbar amphotericin B (TRAMB) injections were given. Posaconazole was used in step-down therapy in a dose of 300 mg BD for two days, followed by 300 mg OD for three months with monitoring of LFT.

Surgical management was performed by a multidisciplinary team, including otorhinolaryngology, ophthalmology, neurosurgery, oral maxillofacial surgery, and plastic surgery teams, depending on the requirement for the case. Our approach was purely endoscopic in most cases. Endoscopic modified Denker’s removal of the middle and inferior turbinate, wide middle meatus antrostomy, ethmoidectomy, sphenoidotomy, frontal sinusotomy, medial/inferior partial maxillectomy, orbital decompression by removal of medial and inferior wall, clearance of ITF and PPF, orbital exenteration, craniotomy, and removal of the abscess were included as part of the surgical approach depending on the needs of the case.

During follow-up, nasal endoscopy and suction clearance were performed every 15 days. Follow-up MRI was performed after 1 and 3 months of posaconazole course or earlier if patients developed any new symptoms. Two patients underwent revision surgery. One developed palatal necrosis, and the other developed a lacrimal abscess. Patients are still under close follow-up for any symptoms and blood sugar monitoring.

### 4.5. Prognosis

The prognosis of invasive fungal sinusitis is poor, with reported mortality rates ranging from 33.3% to close to 100% in disseminated infections even if aggressive debridement and intravenous antifungal agents are utilized [6,33]. Orbital and intracranial extension are associated with increased risk of morbidity and mortality [32]. A study by Turner et al. showed a mortality rate of 50.3% in non-COVID-19 invasive fungal sinusitis. In the current scenario, multicenter studies performed by Singh et al. and Sen et al. show mortalities of 31% and 14%, respectively, in COVID-19-associated invasive fungal sinusitis. Our study’s overall mortality rate is 20.1% (25/124) and 39.1% (9/23) in intracranial cases. Six out of seven patients had large intracranial abscesses and underwent craniotomy and abscess drainage, out of which 50% of patients survived. One patient did not consent to surgery. Two patients with multiple small parietal abscesses and one small temporal lobe abscess were managed nonoperatively. All three survived in the follow-up period and are doing well. Thus, the recovery of the immune system, which is severely compromised in COVID-19, a high index of suspicion in post-COVID-19 patients, early diagnosis, aggressive surgical debridement, and use of antifungals lead to a better survival rate. Neurosurgery intervention should be performed only in patients with large intracranial abscesses. Small abscesses, either single or multiple, and limited intracranial extension can be managed conservatively with antifungals along with PNS debridement.

### 4.6. Limitations

This study is associated with certain drawbacks. First, it was a retrospective analysis. A prospective study with this aspect could reveal more about the epidemiology and risk factors leading to this condition. Secondly, this study was hospital-based in a tertiary referral center, where most cases were referred from primary and secondary treatment centers from all over the state.

## 5. Conclusions

COVID-19 patients, especially high-risk ones, must be followed up by ENT after recovery. COVID-19-related aggressive inflammatory response, uncontrolled glycemic level, and rampant use of steroids are the most important predisposing factors in developing COVID-19-associated invasive fungal sinusitis. A high index of clinical suspicion, diagnosis at an early stage, inflammatory marker monitoring, strict glycemic control, aggressive surgical debridement, and use of antifungal agents are essential for a successful outcome. Finally, to prevent such a dangerous and costly disease, stricter policies have to be implemented and followed to prevent injudicious antibiotic and steroid use. Over-the-counter sale of these drugs should be prohibited, and doctors should strive to follow national and internationally set protocols for their use.

## Figures and Tables

**Table 1 jof-08-00223-t001:** Distribution of the demographic, treatment, inflammatory markers, respiratory support, and outcomes of COVID-19 patients (N = 124).

Variables	Subgroups	Number (%)
COVID-19 status	Positive	20 (16.1%)
	Negative	104 (83.9%)
Sex	Male	87 (70.2%)
	Female	37 (29.8%)
Isolation place	Hospital	66 (53.2%)
	Home	58 (46.8%)
Antibiotics	Yes	91/124 (73.4%)
Antibiotics (intravenous)	Yes	48/91 (52.7%)
Steroid	Yes	90 (72.6%)
Tocilizumab	Yes	3 (2.4%)
Death/cured	Death	25 (20.1%)
	Cured	99 (79.8%)
Type of respiratory support during treatment	Total	69 (55.6%)
	Non-rebreather mask (NRBM)	9 (13%)
		44 (63.8%)
	Bilevel positive airway pressure (BiPAP)	9 (13%)
	Mechanical	7 (10.1%)
Support at the time of admission	Room air	100 (80.6%)
	Ventilator	12 (9.7%)
	Oxygen	12 (9.7%)
Fungal element	Only aspergillus	5 (4%)
	Only mucor	102 (82.2%)
	Both aspergillus and mucor	16 (12.9%)
	Hyalohyphomycosis	1 (0.8%)
Primary antifungal	Amphotericin B	116 (93.5%)
	Isavuconazole	8 (6.5%)
Age (years) #	51.7 ± 11.54 [52 (28, 82)]	
Duration b/w COVID-19 and mucor (days) #	35.4 ± 37.15 [26 (6, 292)]	
Steroid use duration (days) #	11.3 ± 6.6 [10 (3, 45)]	
Duration O2 support (days) #	9.5 ± 8.13 [7 (0, 45)]	
Ferritin #	1780.3 ± 7885.63 [576 (1, 85, 300)]	
HbA1c #	9.1 ± 2.26 [9 (5.2, 14)]	

# Presented as mean ± SD [median (minimum–maximum)].

**Table 2 jof-08-00223-t002:** Clinical symptoms, site, and comorbidities among COVID-19 patients (N = 124).

Variables	Number (%)
Headache	68 (54.8%)
Facial swelling or periorbital swelling	50 (40.3%)
Vision loss/decreased vision	30 (24.2%)
Facial numbness	21 (16.9%)
Nasal discharge	20 (16.1%)
Loose tooth	17 (13.7%)
Tooth pain	16 (12.9%)
Ptosis	11 (8.9%)
Ophthalmoplegia	10 (8.1%)
Proptosis	7 (5.6%)
Skin discoloration	7 (5.6%)
Palatal fistula	6 (4.8%)
Diplopia	4 (3.2%)
Facial palsy	3 (2.4%)
Altered sensorium	2 (1.6%)
**Site of involvement**	
Maxillary	90 (72.6%)
Ethmoids	87 (70.2%)
Nasal cavity	77 (62.1%)
Sphenoid	56 (45.2%)
PPF	46 (37.1%)
Orbit	44 (35.5%)
Palate	41 (33.1%)
ITF	38 (30.6%)
Skull base	30 (24.2%)
Frontal	25 (20.2%)
Intracranial	12 (9.7%)
Skin	9 (7.3%)
**Comorbidities**	
DM (Total)	104 (83.9%)
DM (new onset)	9 (8.6%)
DM current status (uncontrolled)	88 (84.6%)
HTN	38 (30.6%)
CKD/AKD	16 (12.9%)
CAD	8 (6.5%)
GBS	2 (1.6%)
CLD	5 (4%)
Renal transplant	5 (4%)
Hypothyroidism	5 (4%)

PPF, pterygopalatine fossa; ITF, infratemporal fossa; DM, diabetes mellitus; HTN, hypertension; CKD, chronic kidney disease; AKD, acute kidney disease; CAD, coronary artery disease; GBS, Guillain–Barré syndrome; CLD, chronic liver disease.

**Table 3 jof-08-00223-t003:** Predictors of the invasive fungal sinusitis in propensity score-matched data for age, comorbidity, and respiratory supports.

Variables #	Univariate Analysis			Multivariate Analysis $		
	Cases	Control	*p*-Value	AOR	95% CI	*p*-Value
Serum ferritin	576 (276, 1087)	244 (105, 458)	<0.001	1.001	1.001,1.002	0.041
HbA1c	9.1 (7.6, 10.9)	7.7 (6.8, 8.8)	<0.001	1.281	1.05,1.57	0.016
Duration	10 (7, 15)	7 (7, 10)	<0.001	1.12	1.02,1.23	0.014
Steroid (yes)	90 (72%)	39 (31.5%)	<0.001	-	-	-

# Data presented as median (interquartile range), compared by Mann–Whitney U-test. Chi-square test was used to compare the proportion of steroid patients; $ Multivariate binary logistic regression was used to estimate adjusted odds ratio (AOR); COVID-19 patients with and without invasive fungal sinusitis were considered as cases and controls, respectively.

## Data Availability

The data presented in the study are available on request from the corresponding author.

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
