# Peer review of "The Spectrum of Invasive Fungal Sinusitis in COVID-19 Patients: Experience from a Tertiary Care Referral Center in Northern India"

_jof, 2022, doi:10.3390/jof8030223_

Round 1

Reviewer 1 Report

Dear Editor/Author,

Thank you for this opportunity to review this work. The article entitled “The spectrum of invasive fungal sinusitis in COVID-19 patients: A experience from a tertiary care referral center of northern India” is interesting and timely manner. The novelty is good. The title and abstract are matched with the rest of the article. The methodology of the study is valid, reliable, and defined appropriately. The data are presented in an appropriate way. However, the manuscript needs to update the information and I think that the below observations should be addressed before reaching to final decision.

  1. An article published in Journal of Medical Virology regarding black fungus infection in South Asian countries during the COVID-19 pandemic. This is an important article in the same field and discussion of this article would improve the background of this manuscript. Pls see- https://doi.org/10.1002/jmv.27207
  2. Another article on mucormycosis or black fungus in India during covid-19 pandemic with associated risk factors and preventive measures (https://doi.org/10.1016/j.puhip.2021.100153). The authors are asked to discuss this article to improve and update the information.
  3. The authors are advised to mention the limitation of the present study as separate paragraph.
  4. What are the recommendations that the authors have made for the poor and developing countries to reduce the indiscriminate use of steroids and antibiotics during COVID-19 pandemics?
  5. Information and discussion on how earlier influenza pandemics were brought under control; I hope that the information on how similar immunizations or different viral variants were contributed to control the pandemic also helps to improve this article.

I think that most of these issues can be addressed and will help improve the manuscript.

Author Response

Greetings, we are grateful for reviewing our paper and i hope we made required changes as per queries, kind regards

Response to Reviewer 1 Comments

Point 1: An article published in Journal of Medical Virology regarding black fungus infection in South Asian countries during the COVID-19 pandemic. This is an important article in the same field and discussion of this article would improve the background of this manuscript. Pls see- https://doi.org/10.1002/jmv.27207

Response 1: Thank you for pointing out the said letter to editor. This is an interesting and valid point to be discussed and further studied. However to include it as one of points in introduction and discussion we need to back it up with data and analysis rather than news articles and other quoted letter to editors. We would definetly consider further review with this aspect and plan a research analysis so as to proove this point to the policy makers.

Point 2 Another article on mucormycosis or black fungus in India during covid-19 pandemic with associated risk factors and preventive measures (https://doi.org/10.1016/j.puhip.2021.100153). The authors are asked to discuss this article to improve and update the information.

Response 2: As said previously, thank you for pointing out these important letter to editors. However we would like to further back these up with valid research analysis so these points can be irrefutable, without the bias involved with quoted news articles and speculative articles. However we have mentioned the said point in discussion and quoted the article by CDC on mucormycosis.[reference 31]

Point 3 The authors are advised to mention the limitation of the present study as separate paragraph

The limitations have been mentioned as a separate paragraph at the end of the discussion. Section 4.6 were added.

Point 4. What are the recommendations that the authors have made for the poor and developing countries to reduce the indiscriminate use of steroids and antibiotics during COVID-19 pandemics?

These recommendations have been added to the conclusion of the artice.

Point 5. Information and discussion on how earlier influenza pandemics were brought under control; I hope that the information on how similar immunizations or different viral variants were contributed to control the pandemic also helps to improve this article.

Thank you for pointing out this valuable point. However we are pursuing another research in our institute on the covid pandemic and effect of vaccine on control of this pandemic where this point has been extensively studied and mentioned. Thus due to repetition of the data and discussion points we have refrained from commenting on control of covid pandemic in this article and focussed on post covid fungal sinusitis.

Reviewer 2 Report

Thank you for the opportunity to review the manuscript intitled “The spectrum of invasive fungal sinusitis in COVID-19 patients: A experience from a tertiary care referral center of northern India”.

In this paper, the authors conducted a retrospective observational study of COVID-19 patients with invasive fungal disease of the paranasal sinuses, in a tertiary care referral center in India, between April 2021 and September 2021. To assess factors associated with invasive fungal sinusitis in COVID-19 patients and prognosis, patients were matched (on age, sex, respiratory support, and comorbidities) with a cohort of COVID-19 patients without invasive fungal sinusitis.

The main findings are:

  • COVID-19 patients with invasive fungal sinusitis were mainly male, were 51.7±11.54 years old (range 28–82) and had comorbidities such as diabetes mellitus (83.9%), hypertension 38 (30.6%), or kidney disease 16 (12.9%).
  • 2% of COVID-19 patients with invasive fungal sinusitis had mucor, 16.9% aspergillus and 12.9% had both
  • Invasive fungal sinusitis is a late complication (35.4±37.2 days after COVID diagnosis). Around a half of patients were at hospital for COVID-19 treatment, whereas the others were discharged at home post-COVID-19 at the time of sinusitis diagnosis.
  • Almost all patient (72.6%) received corticosteroids for COVID-19 for mean duration of 11.3±6.6 days, whereas about a half (55.64%) required oxygen support.
  • By multivariate analysis, serum ferritin (AOR: 1.001, p=0.041), HbA1c (AOR: 28, p<0.001) and duration of steroid use (AOR: 1.12, p=0.014) were significant and independent risk factors for invasive fungal sinusitis.
  • Urgent surgical debridement and antifungal therapy is mandatory, and mortality rate remains high at 20%

This paper is very interesting, findings are well described and the manuscript is appropriately organized.

I have the following comments:

  • It could be very interesting and informative to report the total number of COVID‑19 patients admitted at SGPGIMS Lucknow tertiary referral center, India, over the study period in order to assess or approximate the incidence of invasive fungal sinusitis in COVID-19 patients
  • The authors should insist more on the fact that invasive fungal sinusitis overwhelmingly occur in discharged patients. It is an important difference with other invasive fungal infection reported previously in COVID-19 patients which occur more frequently in those with critical illness.
  • The authors have to clearly explain why a large majority (73.4%) of patient with invasive fungal sinusitis have been treated with antibiotics (IV and oral)
  • It is surprising that 72.6% of patients received corticosteroids for COVID-19 whereas about only 55.6% required oxygen support. In the same line the median duration was 11.3±6.6 days. The WHO guidelines and large RCT (RECOVERY) recommend corticosteroids for maximum 10 days or until discharge and only in those with oxygen support… Please justify
  • The timing of ferritin measurement was not clear… At the time initial hospitalization? At the time of invasive fungal sinusitis diagnosis? Please clarify.
  • It is not clear for me, if the authors have integrated all patients with invasive fungal sinusitis in the multivariate analysis or only those mucormycosis. Factors associated with mucormycosis ans those associated with aspergillosis are reported to be slightly different in the literature of COVID-19 associated mucormycosis (CAM) and COVID-19 associated pulmonary aspergillosis (CAPA). Justify.
  • India is a place of high prevalence for mucormycosis. Please has a statement to limit the generalization of the study results to other countries…
  • Please change in the title: a experience  an experience
  • Percentages are sometimes reported with two decimal (XX.XX%) or one (XX.X%). Please change in XX.X%.
  • Please had space between text and citation: example [1]
  • Add “%” after 53.2, after 46.8 and 47.3 in the text line 95, p2 and line 96 p3 and line 98 p3, respectively
  • The process chosen to selected the variable into the multivariable analysis is not clearly explained
  • Remove the point “.” line 216 p6
  • Please chose (Table X) or [Table X] according to the recommendations for authors and uniformize (see line 106, 118, 138, 159, 223…)
  • A “T” is missing line 240, p7
  • Limit abbreviations and explain each abbreviation used for clarity pf the manuscript

One more time, thank you for the opportunity to review your manuscript.

Author Response

Greetings , we are grateful your wonderful comments and thorough analysis of our paper, i hope we made required changes as per queries, kind regards

Response to Reviewer 2 Comments

Point 1: It could be very interesting and informative to report the total number of COVID‑19 patients admitted at SGPGIMS Lucknow tertiary referral center, India, over the study period in order to assess or approximate the incidence of invasive fungal sinusitis in COVID-19 patients

Response 1: the main reason to exclude the total number of covid patients in our centre was that most of the patients who have been referred to us with mucormycosis were treated at different covid centres from nearby region.

Point 2: The authors should insist more on the fact that invasive fungal sinusitis overwhelmingly occur in discharged patients. It is an important difference with other invasive fungal infection reported previously in COVID-19 patients which occur more frequently in those with critical illness.

Response: thank you for pointing out this wonderful point. We have included the said changes in the introduction.

Point 3: It is surprising that 72.6% of patients received corticosteroids for COVID-19 whereas about only 55.6% required oxygen support. In the same line the median duration was 11.3±6.6 days. The WHO guidelines and large RCT (RECOVERY) recommend corticosteroids for maximum 10 days or until discharge and only in those with oxygen support… Please justify

Response: the main issue with addressing this point is that not all patients have been treated for covid at our centre and hence its difficult to comment the treatment protocol used at different centres.

Point 4: It is not clear for me, if the authors have integrated all patients with invasive fungal sinusitis in the multivariate analysis or only those mucormycosis. Factors associated with mucormycosis ans those associated with aspergillosis are reported to be slightly different in the literature of COVID-19 associated mucormycosis (CAM) and COVID-19 associated pulmonary aspergillosis (CAPA). Justify.

Response: We have included the patients of all types of fungal pathogens causing invasive fungal sinusitis so as to get better assessment of the available data.

Point 5

Please change in the title: a experience  an experience

Percentages are sometimes reported with two decimal (XX.XX%) or one (XX.X%). Please change in XX.X%.

Please had space between text and citation: example [1]

Add “%” after 53.2, after 46.8 and 47.3 in the text line 95, p2 and line 96 p3 and line 98 p3, respectively

The process chosen to selected the variable into the multivariable analysis is not clearly explained

Remove the point “.” line 216 p6

Please chose (Table X) or [Table X] according to the recommendations for authors and uniformize (see line 106, 118, 138, 159, 223…)

A “T” is missing line 240, p7

Limit abbreviations and explain each abbreviation used for clarity pf the manuscript

Response: thank you pointing out these mistakes and we apologise for not correcting it ourself.

Point 6: India is a place of high prevalence for mucormycosis. Please has a statement to limit the generalization of the study results to other countries

The said statement has been made. Thank you